

# Metabolomic and physiological analysis of bud differentiation in dense apple (*Malus × domestica* Borkh.) orchards following thinning and reshaping

Zehua Yang[1], Tianli Guo[2], Junqiang Niu[1], Xiaoning Yin[1] and Ming Ma[1]

[1] Institute of Forestry, Fruits and Floriculture, Gansu Academy of Agricultural Sciences, Lanzhou, Gansu, China
[2] College of Agriculture, Guangxi University, Nanning, Guangxi, China

## ABSTRACT

**Background**. In China, a majority of apple orchards were initially planted under arboriculture. The branch number increases, causing a deterioration of the canopy light conditions as the trees age. This phenomenon blocks bud differentiation, resulting in yield decrease and deterioration of fruit quality. Thinning and reshaping are effective strategies for addressing these issues. However, the impact of thinning and reshaping remains unclear. This study analyzed the physiological and metabolic aspects of flower bud differentiation following thinning and reshaping of an overcrowded orchard.

**Methods**. Physiological and metabolomic analyses were conducted on terminal flower bud samples collected during early (T1) and late (T2) bud differentiation following thinning and reshaping alongside controls CK1 and CK2.

**Results**. Sucrose, glucose, fructose, and sorbitol contents in T1 and T2 groups were significantly higher than in CK1 and CK2. Metabolomic analysis revealed significant differences: 60 metabolites (32 up-regulated, 28 down-regulated) in T1 *vs* CK1 and 51 metabolites (26 up-regulated, 25 down-regulated) in T2 *vs* CK2. KEGG enrichment analysis revealed that the biosynthesis of secondary metabolites was the most enriched pathway, with the crucial compounds associated with flower bud differentiation. There was an accumulation of coumarin (C05851), eriodictyol (C05631), and histidine (C00135) during both the early and late stages of flower bud differentiation, underscoring their promoting role. In T1 *vs* CK1, 2-isopropylmalic acid (C02504), (+)-catechin (C09727), and trans-ferulic acid (C01494) were significantly up-regulated, highlighting their potential as metabolic activators. Conversely, 3',5'-dimethoxy-3,5,7,4'-tetrahydroxyflavone (C11620) and 4-hydroxyphenylpyruvic acid (C01179) decreased, implying divergent regulatory mechanisms. Temporal specificity emerged in T2 *vs* CK2, with significant upregulation of myricetin (C10107) and isopimpinellin (C02162). In contrast, umbelliferone (C09315) and trans-caffeic acid (C01197) exhibited opposing trends. Moreover, jasmonic acid (C08491) and 6-phosphogluconic acid (C00345) increased sharply in T2, while trans-traumatic acid (C16308) declined.

**Conclusions**. Thinning and reshaping improved light penetration and increased the accumulation of nutrients and secondary metabolites. Sucrose, glucose, fructose, and sorbitol contents and the relative levels of coumarin, eriodictyol, and histidine increased during early and late flower bud differentiation stages, which suggested

Corresponding author
Junqiang Niu, niujq222@sina.com

their positive roles during these stages. 2-isopropylmalic acid, (+)-catechin, and trans-ferulic acid exhibited metabolic activation potential, while 3',5'-dimethoxyflavone and 4-hydroxyphenylpyruvic acid were distinctly regulated in T1 *vs* CK1. Myricetin, isopimpinellin, jasmonic acid, and 6-phosphogluconic acid exhibited activation potential, while trans-traumatic acid was distinctly regulated in T2 *vs* CK2. Metabolic changes and pathway-specific activation/inhibition patterns underlying flower bud differentiation exhibited a direct-indirect regulatory network in secondary metabolism.

## INTRODUCTION

Apple (*Malus × domestica* Borkh.) is one of China's most important economic crops, with its planting area and annual production accounting for 43.33% and 50.96% of the world's total, respectively (*FAO, 2023*). The differentiation of flower buds is closely linked to the yield and quality of apple fruits, marking a critical transition from vegetative to reproductive growth. This process is regulated by a complex interplay of internal factors, such as hormones and carbohydrates, and external conditions, including light and temperature, which collectively ensure successful flowering (*Lee & Lee, 2010*; *Turnbull, 2011*; *Kurokura et al., 2013*). Most apple orchards are planted using a high-density cultivation system with standard-sized trees during the early stages of orchard establishment. Although this approach initially yields increased benefits, orchard density increases as the trees age, resulting in overcrowded canopies and deteriorating light conditions. This phenomenon hinders flower bud differentiation, resulting in a decline in fruit yield and quality (*Griffin, Ranney & Pharr, 2004*), which severely restricts the development of the apple industry. The restructuring of overgrown, monotonous, and standard apple orchards, as well as modifying the tree structure, have become the main hurdle in cultivating older apple orchards.

Thinning and reshaping overcrowded apple orchards are the main techniques for modifying and restructuring the mature, densely planted standard apple orchards. These techniques can quickly enhance orchard ventilation and light conditions, optimizing tree structure, flower bud differentiation, and internal and external fruit quality. Some studies suggest that significant improvements occur in flower bud differentiation, flowering rate, quality of flower buds, distribution of flower buds and fruits in the canopy, and fruit-setting rate in the third year after thinning and reshaping old orchards (*Niu et al., 2020*). Despite the well-established benefits of thinning and reshaping on apple flower bud differentiation, the underlying changes in carbohydrate metabolism and the associated metabolic mechanisms during this process remain unexplored. The molecular mechanisms in question should thus be explored through further studies.

Carbohydrates play a crucial role in the transition of the apical meristem from vegetative to reproductive development (*Bernier & Périlleux, 2005*). Of note, the creation of flower organs necessitates the provision of a large amount of carbohydrates (*Chmielewski &*

*Götz, 2022*). Previous studies postulate that the differentiation of flower buds is caused by elevated amounts of sucrose, glucose, fructose, and sorbitol (*Shang et al., 2022*). On one hand, sucrose serves as an inducer of flowering, acting as an initial signal (*Xing et al., 2015*). On the other hand, glucose is responsible for the early growth of organs by osmotic expansion in the neighboring dividing cells (*Smeekens et al., 2010*). Fructose regulates flower bud differentiation by affecting the polar transport of auxins (*Eveland & Jackson, 2012*). Sorbitol is the primary photosynthetic product and translocated carbohydrate that is largely converted into fructose by sorbitol dehydrogenase (*Wu et al., 2010*). Notably, sorbitol levels peak during the early stages of flower bud development. In apples, sucrose plays a pivotal signaling role in flower bud differentiation, regulating the expression of flowering-related genes, such as *SUPPRESSOR OF CONSTANS 1* (*SOC1*) and *AGAMOUS-like 24* (*AGL24*) (*Xing et al., 2015*). In loquat (*Eriobotrya japonica* Lindl.), sorbitol levels increase during flower bud differentiation (*Xu et al., 2023*). To date, only a few studies have investigated whether thinning and remodeling can increase carbohydrate content, thereby affecting flower bud differentiation.

Flavonoids have been widely studied to check their developmental regulation capabilities in plant reproduction, which in turn affect a flower's main processes, such as flower formation and blooming, growth of pollen tubes, and the development and maturation of fruits and seeds (*Taylor & Grotewold, 2005*; *Broun, 2005*). Numerous recent studies suggest that the addition of certain flavonoid compounds can promote full flowering stages, reproductive development, and fruit set rates (*Yang et al., 2020*; *Yang et al., 2021*). Noteworthy, the eriodictyol content in the flower buds of *Chrysanthemum morifolium* significantly increases when flower bud differentiation undergoes different stages of waterlogging stress (*Wang et al., 2019*).

The phenylpropanoid biosynthesis pathway is greatly responsible for flower bud differentiation (*Fan et al., 2018*). Flower bud formation is promoted by metabolites involved in this pathway, which act as plant regulators (*Dudareva, Pichersky & Gershenzon, 2004*). These metabolites serve as vital intermediaries in plants' response to abiotic (*e.g.*, light and minerals in the soil) and biotic (*e.g.*, pests) stimuli (*Dudareva, Pichersky & Gershenzon, 2004*). They are also endowed with remarkable antioxidant and free radical scavenging properties (*Vogt, 2010*). *Xing et al. (2015)*, the first researcher to investigate the possible modes of action of phenolic compounds and proline in regulating flower bud growth and initiation in fruiting trees of *Myrica rubra*, postulates that the phenylpropanoid metabolism pathway is involved in the process that leads to flower bud differentiation.

The merging of physiological metabolism and metabolomics enables a systematic and thorough study of molecular functions and regulatory mechanisms at both physiological and metabolite transformation levels (*Tian et al., 2023*). This "physiological-metabolomics" method has been employed to investigate the impact of environmental changes on physiological parameters and variations in metabolites (*Ju et al., 2023*). In this study, the dense canopy of the 'Changfu No. 2' Fuji orchard was thinned and reshaped to create space for lighting and ventilation between trees. The influence of thinning and reshaping on flower bud differentiation was subsequently studied by amalgamating physiological metabolism and metabolomics methods. This study elucidated the dynamic

patterns of regulatory substances and established a network mechanism to decipher the metabolic interaction patterns governing flower bud differentiation through canopy thinning and architectural modification.

## MATERIALS & METHODS

### Plant material and treatments

The experiment was conducted at the National Apple Industry Technology System Pingliang Apple Comprehensive Experiment Station, located in Jingning Demonstration County, Chengchuan Town, China (35°24′16″N, 105°47′15″E) in 2022. The experimental trees were 15-year-old 'Changfu No. 2' Fuji apple trees grafted onto *Malus robusta* rootstocks. The trees were planted at a spacing of 4 m × 3 m, leading to a density of 834 trees per hectare. The trees were trained in an improved spindle shape with heights ranging between 4.2 and 4.6 m, a trunk height of 60 to 75 cm, and six to nine main branches angled at between 70° and 80°.

The thinning and reshaping restructuring experiment was initiated in mid-February 2022 in the experimental orchard. A completely randomized block design with thinning and restructuring as the single experimental factor, consisting of two treatments: T (thinning with restructuring) and CK (control without thinning or restructuring), was adopted. In the T treatment, alternate trees were removed to adjust the original planting density from 4 m × 3 m to 4 m × 6 m spacing. A comprehensive restructuring of the canopy was achieved through the selective removal of large branches, elevation of trunk height to 100–110 cm, and elimination of the central leader, resulting in tree heights of 310–340 cm. Four to six healthy, well-distributed scaffold branches were retained as permanent structural elements with branch angles systematically adjusted to 90°–100°. Standard Fuji apple pruning protocols were followed to achieve a total pruning intensity of approximately 30%. The control group (CK) maintained the original planting density and canopy architecture throughout the experiment. Each 500 m$^2$ treatment plot contained three replicates with five representative trees of moderate vigor. There were no visible pests or disease symptoms in any of the replicates. Sampling was conducted during critical phenological stages on June 22 (early physiological differentiation) and August 22, 2022 (late physiological differentiation). Sampling entailed collecting terminal buds from medium- and short-shoots (length < 15.0 cm) in the mid-lower canopy region, with strict adherence to uniformity criteria for bud size and developmental stage. Samples were individually wrapped in pre-labeled aluminum foil immediately after collection, flash-frozen in liquid nitrogen, and stored at −80 °C to preserve biochemical integrity for subsequent analysis.

### Determination of carbohydrate content

The contents of sucrose, glucose, fructose, and sorbitol in the treatment and control groups were measured using the respective commercially available assay kits. These kits were purchased from Merck KGaA (Darmstadt, Germany) with the following product reference numbers: SCA20 for the sucrose assay kit, MAK013 for the glucose assay kit, FA20 for the fructose assay kit, and PHR1006 for the sorbitol assay kit.

## Metabolomics analysis

Apple blossom buds (100 mg) were homogenized in 300 μL of 50% methanol and incubated at 4 °C overnight. The mixture was then centrifuged at 13,000 rpm for 20 min at the same temperature. The supernatant was then carefully collected and concentrated using a vacuum freeze dryer. The dried residue was diluted in 100 μL of methanol for LC/MS analysis. A Nexera X2 system (Shimadzu, Kyoto, Japan) was coupled with an AB Sciex Triple TOF 5600 system (AB Sciex, Framingham, MA, USA) for LC/MS analysis, resulting in a substantial amount of mass spectrometry data.

## Non-targeted metabolomic analysis

Frozen samples ($-80$ °C) were thawed on ice, homogenized with 400 μL of methanol/water (7:3, v/v) containing an internal standard (20 mg sample), and vortexed for 3 min. The homogenate was subsequently sonicated in ice for 10 min, vortexed for 1 min, and incubated at $-20$ ° C for 30 min. The mixture was then centrifuged (12,000 rpm, 10 min, 4 °C) to obtain the supernatant. The supernatant was re-centrifuged (12,000 rpm, 3 min, 4 °C), and 200 μL of the clarified supernatant was subjected to LC-MS analysis. The settings of the positive ion mode were: separation on a Waters ACQUITY Premier HSS T3 column (1.8 μm, 2.1 × 100 mm) using 0.1% formic acid in water (A) and 0.1% formic acid in acetonitrile (B) with the gradient set as follows: 5% B → 20% B (2 min), → 60% B (3 min), → 99% B (1 min), hold 1.5 min, → 5% B (0.1 min), and equilibrate for 2.4 min. The column temperature was 40 °C, the flow rate was 0.4 mL/min, and the injection volume was 4 μL. The settings of the negative ion mode were similar to those of the positive-ion mode except for ionization polarity.

## Data processing and bioinformatics analysis

Signal intensity information for each compound across different samples was extracted using the XCMS software and processed using the metaX software for quality control. The workflow involved removing low-quality peaks (those missing in more than 50% of QC samples or missing in over 80% of actual samples), imputing missing values using the KNN (R (impute) software, Version 1.56.0, default parameters), and data normalization through probabilistic quotient normalization (PQN) followed by QC-robust spline batch correction (QC-RSC). The corrected and filtered peaks were subjected to metabolite identification by searching against the laboratory's in-house database and integrating public databases (HMDB, https://hmdb.ca/, and KEGG, https://www.kegg.jp/). The annotated metabolites were then mapped to the KEGG Pathway database (http://www.kegg.jp/kegg/pathway.html). Unsupervised principal component analysis (PCA) was conducted using the prcomp function in R (http://www.r-project.org) for dimensionality reduction and pattern recognition. Orthogonal partial least squares-discriminant analysis (OPLS-DA) was conducted using the R package MetaboAnalystR to extract variable importance in projection (VIP) values and generate model diagnostics. Differential metabolites between groups were identified using the following criteria: VIP > 1 (from OPLS-DA), fold-change thresholds ($\geq 2$ or $\leq 0.5$), and statistical significance of $P < 0.05$ (two-tailed Student's $t$-test).

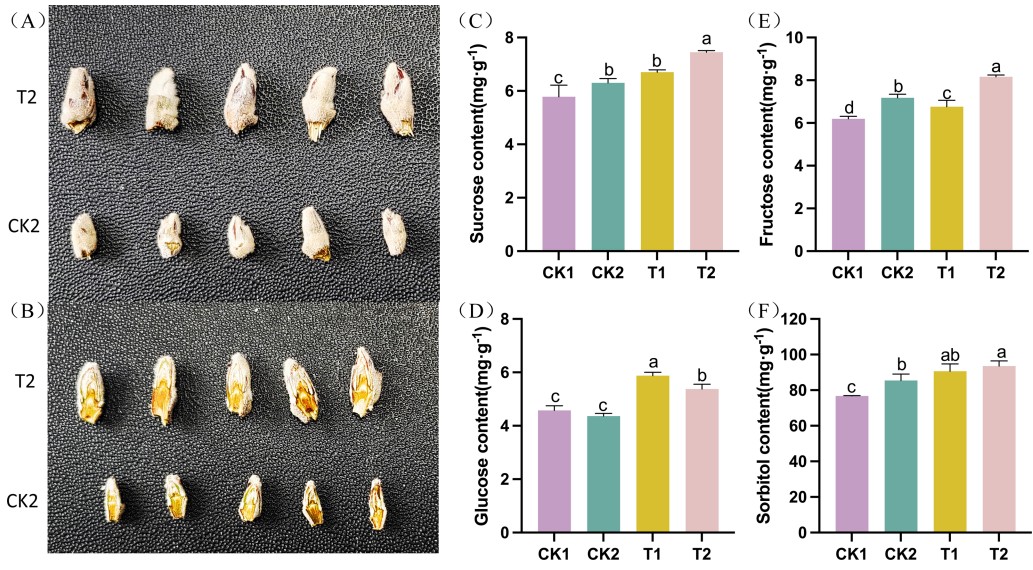

**Figure 1** **Phenotypic and physiological characteristics of flower bud differentiation after thinning and reshaping in the different treatments.** (A) The phenotype of intact apple flower buds and (B) longitudinal section of apple buds in different treatment groups. (C) Bar graph showing the contents of sucrose in the different treatments. (D) Bar graph showing the contents of glucose in the different treatments. (E) Bar graph showing the contents of fructose in the different treatments. (F) Bar graph showing the contents of sorbitol in the different treatments. Note: data is presented as "means ±standard deviation." Significant differences between groups were analyzed using the one-way ANOVA Tukey test ($P < 0.05$) and are denoted by the different letters.

# RESULTS

## Phenotypic and physiological characteristics

The differentiation of apple flower buds was examined following thinning and reshaping treatment. The flower buds were gathered and photographed at the late stage of flower bud differentiation (T2 *vs* CK2). Flower buds in the T2 group were significantly bigger and fuller than those in the CK2 group. They also had more layers in their longitudinal section (Figs. 1A–1B). The levels of carbohydrates (such as sucrose, glucose, fructose, and sorbitol, among others) varied significantly throughout the flower bud differentiation process. The sucrose, sorbitol, fructose, and glucose levels in T1 were significantly higher than those in CK1 during the early stages of flower bud development. Similarly, T2 had significantly higher amounts of sucrose, sorbitol, fructose, and glucose than CK2 during the late stage of flower bud differentiation. These findings suggested that thinning and reshaping positively promote flower bud differentiation during the early and late stages (Figs. 1C–1F).

## Multivariate statistical analysis

Each group had six biological replicates with high consistency to ensure data reliability. Quality control analyses were conducted before the study to ensure the reliability of the detection results and good intra-treatment replicability (Fig. S1). PCA results revealed significant separation of T1 and CK1 samples (Fig. 2A) and T2 and CK2 samples (Fig. 2B), indicating significant differences between treatments. These results highlighted

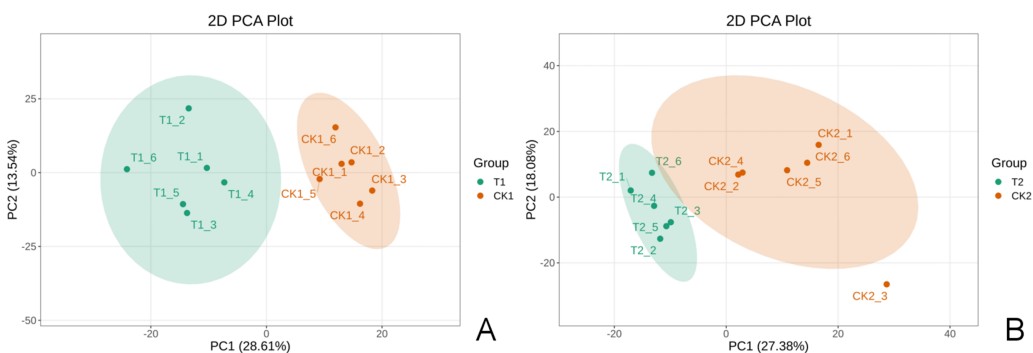

**Figure 2** **The 2D principal component analysis diagram.** A is the T1 *vs* CK1 comparison group and B is the T2 *vs* CK2 comparison group. Note: PC1 represents the first principal component, PC2 represents the second principal component, the percentage represents the interpretation rate of this principal component for the data set. Each point in the figure represents a sample. Samples in the same group are represented by the same color.

the significant metabolic profile differences between T1 and CK1, and between T2 and CK2.

The TIC maps were quite identical, which demonstrated the instrument's strong stability and data accuracy (Fig. S1).

## Metabolic profiling

Metabolomic composition analysis revealed hierarchical characteristics in the metabolite distribution among different experimental groups. Secondary classification of 1,312 metabolites from the T1 *vs* CK1 comparison group revealed the following top five metabolite categories and their proportions: lipids and lipid-like molecules (23.3%), phenylpropanoids and polyketides (20.6%), organic acids and derivatives (10.8%), benzene-ring-containing compounds (10.6%), and heterocyclic compounds (9.4%) (Table S1A). The specific compositional feature of dominant categories in the lipids and lipid-like molecules category was fatty acyls (39.7%) (Table S1B). The phenylpropanoids and polyketides category was dominated by flavonoids (48.1%) and cinnamic acids and derivatives (20.0%) (Table S1C). The organic acids and derivatives category was dominated by carboxylic acids and derivatives (80.9%) (Table S1D). The benzene-ring-containing compounds category was dominated by benzene and substituted derivatives (74.8%) (Table S1E). The heterocyclic compounds category was dominated by imidazopyrimidines (10.6%) and indoles and derivatives (9.8%) (Table S1F). Notably, the top five secondary classifications in the T2 *vs* CK2 comparison group involving 1,306 metabolites remained completely consistent with those of the T1/CK1 group (Table S2A). However, there were distinct differences in subclass distributions. The lipids and lipid-like molecules category was dominated by fatty acyls (39.7%) and terpenoid lipids (30.5%) (Table S2B). The subclass compositions of the other four major categories (Tables S2C–S2F) remained highly consistent with those of the T1/CK1 group, with no significant changes in the proportions of the dominant subclasses, *i.e.,* flavonoids (48.1%), carboxylic acid derivatives (80.9%), benzene-substituted compounds (74.8%), and imidazopyrimidines (10.6%).

## Composition differences of the metabolites

There were 60 metabolites with significantly different expressions (32 up-regulated and 28 down-regulated) in the T1 *vs* CK1 comparison group (Fig. 3A). In contrast, the T2 *vs* CK2 comparison group had 51 metabolites with significantly different expressions (26 up-regulated and 25 down-regulated) (Fig. 3B). The top 10 metabolites with the largest absolute value of Log2FC in each comparison group were selected based on the screening criteria for the drawing of the radar chart. The top 10 metabolites in the T1 *vs* CK1 comparison group based on difference multiple were trifloxystrobin, difenoconazole, pyraclostrobin, propiconazole, 11(12)-Epoxy-5Z,8Z,14Z-eicosatrienoic acid, acetamiprid, hippuric acid, methyl ester, coumarin, 4-Formylbenzonitrile, and spirodiclofen (Fig. 3C). In the T2 *vs* CK2 comparison group, the top 10 metabolites based on difference multiple were trifloxystrobin, difenoconazole, pyraclostrobin, propiconazole, prodione, flusilazole, tebuconazole, m-Hydroxybenzoylecgonine, acetamiprid, and 4-formylbenzonitrile (Fig. 3D).

## KEGG metabolic pathway analysis and screening of metabolites related to flower bud differentiation

KEGG metabolic pathway enrichment analysis ($p < 0.05$, fold change (FC) $\geq 2$ or FC$\leq 0.5$) was done to better understand the mechanism by which thinning and reshaping enhance apple flower bud differentiation.

Notably, biosynthesis of secondary metabolites was the most significant common metabolic pathway in the two comparison groups. Biosynthesis of amino acids of secondary metabolites, flavonoids, flavone and flavonols, biosynthesis of various plant secondary metabolites, phenylpropanoid biosynthesis, and biosynthesis of various alkaloids were significantly enriched in the T1 *vs* CK1 comparison group (Fig. 4A). Eight metabolites were enriched in these pathways. Biosynthesis of amino acids of secondary metabolites, flavonoids, flavone and flavonols, biosynthesis of various plant secondary metabolites, phenylpropanoid biosynthesis, biosynthesis of various alkaloids, alpha-linolenic acid metabolism, plant hormone signal transduction, pentose phosphate pathway, and carbon metabolism were all significantly enriched in the T2 *vs* CK2 comparison group. A total of 10 metabolites were enriched in these pathways (Fig. 4B).

Histidine (C00135) participated in the biosynthesis of amino acids, and eriodictyol (C05631) participated in flavonoid biosynthesis. In contrast, coumarin (C05851) participated in the biosynthesis of various plant secondary metabolites in the two comparison groups (Tables S3A & S3B).

In the T1 *vs* CK1 comparison group (Table S3A), 2-isopropylmalic acid (C02504) and 4-hydroxyphenylpyruvic acid (C01179) were involved in the biosynthesis of various plant secondary metabolites. (+)-catechin (C09727) was involved in flavonoid biosynthesis, 3′, 5′-dimethoxy-3, 5, 7, 4'-tetrahydroxyflavone (C11620) in flavone and flavonol biosynthesis, and trans-ferulic acid (C01494) in phenylpropanoid biosynthesis and biosynthesis of various alkaloids.

Umbelliferone (C09315) and isopimpinellin (C02162) participated in the biosynthesis of various plant secondary metabolites in the T2 *vs* CK2 comparison group (Table S3B), which was also included. Myricetin (C10107) was involved in flavonoid and flavone and flavonol

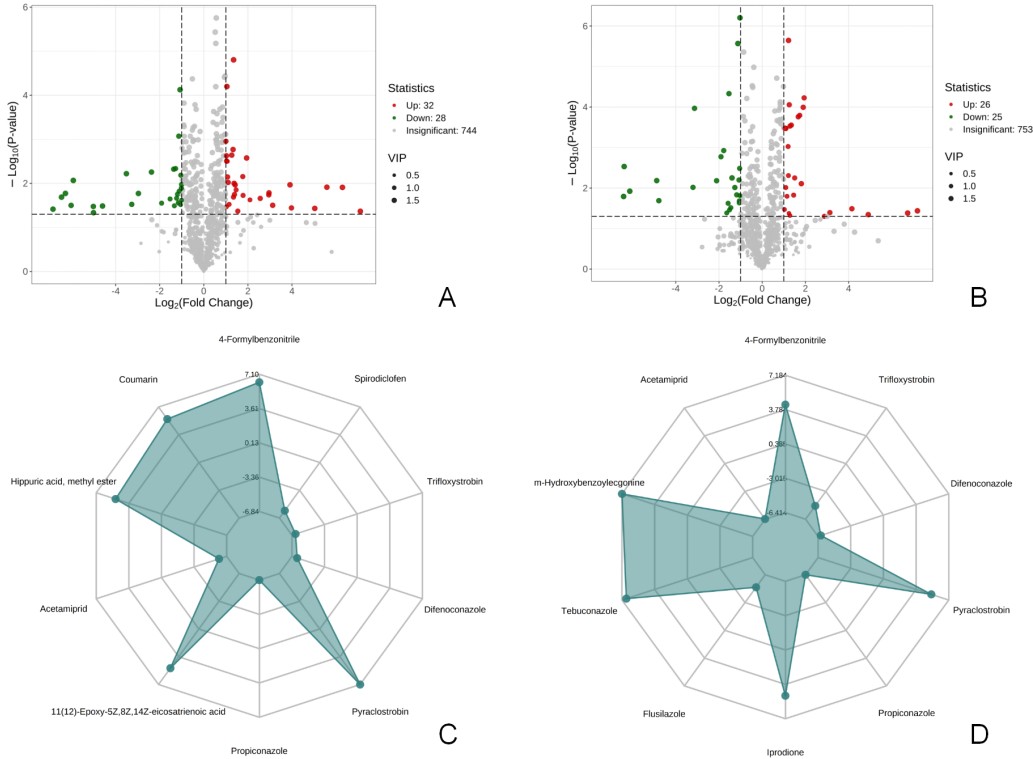

**Figure 3 Volcano map of differential metabolites and r adar chart of the top 10 differential metabolites.** (A) Volcano map of differential metabolites in T1 *vs* CK1 comparison group (B). Volcano map of differential metabolites in T2 *vs* CK2 comparison group Note: Under the VIP + FC + *P*-value screening conditions , the horizontal axis represents the logarithm of the multiple of the relative content difference of a certain metabolite in the two groups of samples (Log2FC). The larger the absolute value of the horizontal axis, the greater the relative content difference of the substance between the two groups of samples. The vertical coordinate represents the level of significance of the difference (-Log10P-value). The size of the dots represents the VIP value. Green dots represent down-regulated differential metabolites, red dots represent up-regulated differential metabolites, and gray represents detected metabolites with no significant difference. (C) Radar chart of the top 10 differential metabolites in T1 *vs* CK1 comparison group with the largest absolute value of Log2FC (D). Radar chart of the top 10 differential metabolites in T2 *vs* CK2 comparison group with the largest absolute value of Log2FC. Note: the grid lines correspond to Log2FC, the green shadow is composed of the Log2FC lines of each substance.

biosynthesis. Trans-caffeic acid (C01197) was involved in phenylpropanoid biosynthesis and biosynthesis of various alkaloids. Jasmonic acid (C08491) and transtraumatic acid (C16308) were involved in alpha-linolenic acid metabolism. Tasmonic acid (C08491) was involved in plant hormone signal transduction. 6-phosphogluconic acid (C00345) was involved in the pentose phosphate pathway and carbon metabolism.

L-histidine, 2-Isopropylmalic acid, eriodictyol, (+)-catechin, coumarin, and trans-Ferulic acid were relatively increased in the T1 *vs* CK1 comparison group (Fig. 5). In contrast, 3′, 5′-Dimethoxy-3, 5, 7, 4′-tetrahydroxyflavone and 4-hydroxyphenylpyruvic acid were relatively reduced in this comparison group. Isopimpinellin, coumarin, trans-caffeic acid, jasmonic acid, eriodictyol, and 6-phosphogluconic acid were relatively increased in the T2 *vs* CK2 comparison group (Fig. 6). In contrast, myricetin, histidine, umbelliferone,

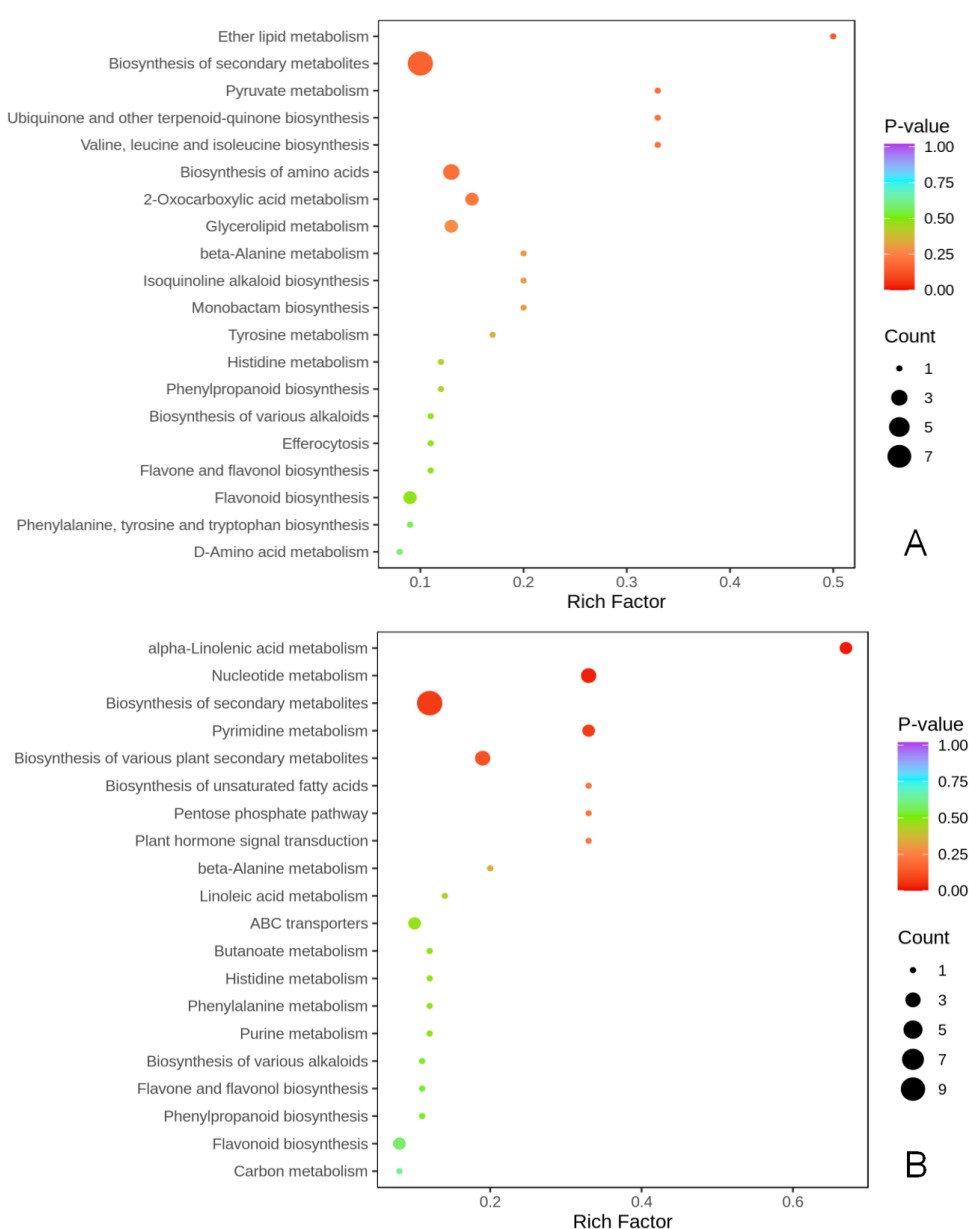

**Figure 4  KEGG enrichment of differential metabolites.** (A) KEGG enrichment of differential metabolites in T1 *vs* CK1 comparison group (B). KEGG enrichment of differential metabolites in T2 *vs* CK2 comparison group. Note: the horizontal axis represents the rich factor corresponding to each path, and the vertical axis is the path name (sorted by *P*-value). The color of the points reflects the size of the *P*-value, and the redder it is, the more significant the enrichment. The size of the dots represents the number of differential metabolites enriched.

and trans-traumatic acid were relatively reduced in this comparison group. Notably, eriodictyol and coumarin exhibited a consistent increase in both comparison groups, indicating that their functions were potentially similar in the process of regulating flower bud differentiation. Histidine exhibited an opposite trend in the two comparison groups,

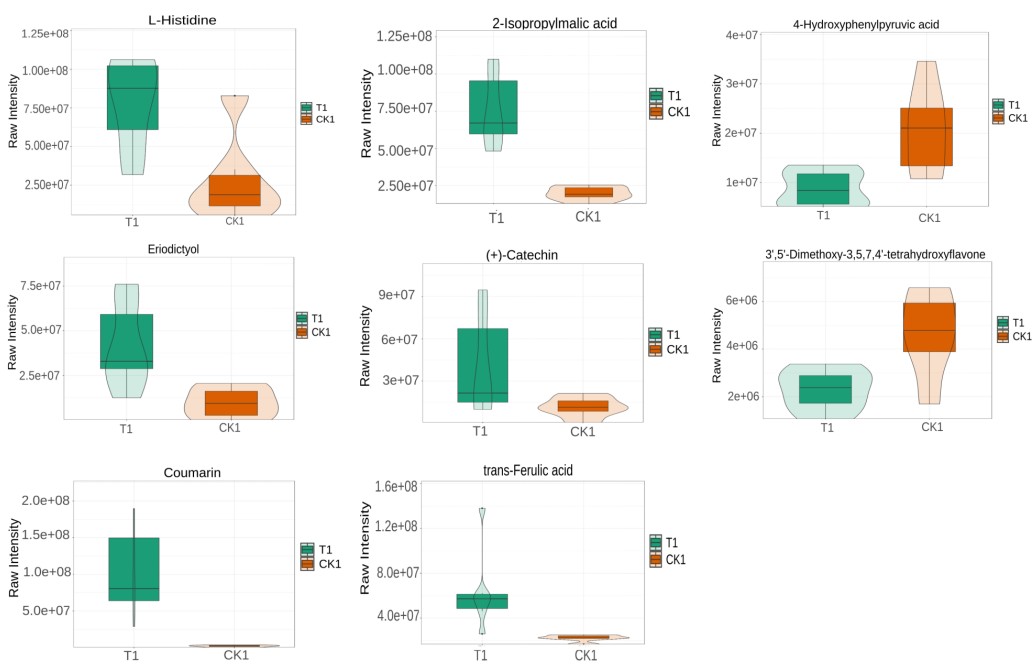

**Figure 5** **Violin diagram of differential metabolites involved in biosynthesis of secondary metabolites pathways in T1 *vs* CK1 comparison group.** Note: the horizontal axis represents the grouping, and the vertical axis represents the relative content of differential metabolites (original peak area).

suggesting that its functions potentially change in the process of regulating flower bud differentiation.

## DISCUSSION

### Flower bud differentiation is closely associated with carbohydrate content

Carbohydrates are essential structural compounds and energy sources in plant growth and development. They play a crucial role in flower bud differentiation. Thinning and reshaping of fruit orchards enhance light intensity, thereby enhancing flowering. Studies postulate that carbohydrates are involved in the flowering process of chrysanthemums under photoperiod induction (*Wang et al., 2010*). Glucose and fructose are directly utilized to provide sufficient nutrients for flower bud differentiation. Notably, their content is positively correlated with the differentiation process. Sorbitol may also be physiologically transformed into fructose and glucose (*Morandi et al., 2008*), thereby promoting flower bud differentiation. Sucrose is the primary product of photosynthesis and is the main form of sugar transport in fruit trees. It also acts as a signaling factor for flowering in plants (*Usenik, Fabčič & Štampar, 2008*). *Xing et al. (2015)* demonstrated that the glucose and fructose content in the terminal flower buds and adjacent leaves of 'Nagafu 2' apples change significantly during the flower induction stage. In the same line, *Barbier et al. (2015)* postulated that sucrose accumulation promotes flower bud differentiation in roses. A study by *Xu et al. (2023)* reported that the sorbitol content significantly increases in

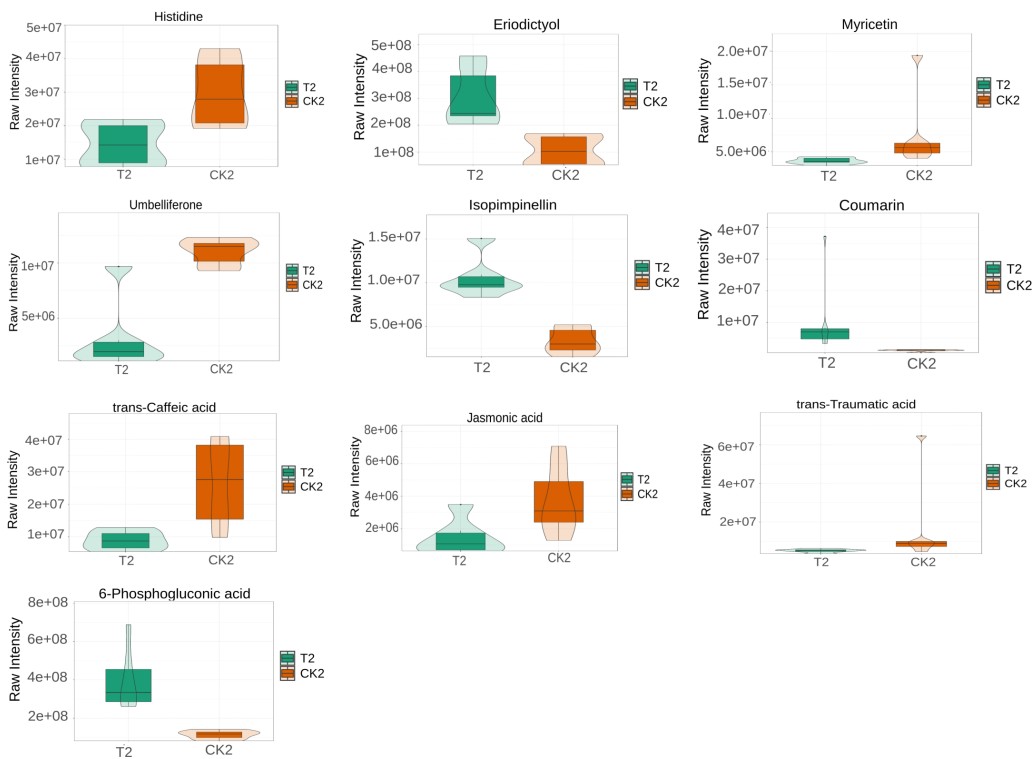

**Figure 6** **Violin diagram of differential metabolites involved in biosynthesis of secondary metabolites pathways in T2 *vs* CK2 comparison group.** Note: the horizontal axis represents the grouping, and the vertical axis represents the relative content of differential metabolites (original peak area).

loquat flower buds during differentiation, with exogenous sorbitol application further promoting flower bud formation. Herein, thinning and reshaping of the apple orchard increased light intensity, thereby increasing the amount of glucose, fructose, sucrose, and sorbitol in both the early and late phases of flower bud differentiation and accelerating the differentiation process.

## Flower bud differentiation is closely associated with secondary metabolites

Plant hormones influence plant germination, flowering, fruiting, dormancy, abscission, and participate in floral regulation (*Du, Yu & Zhou, 2021*; *Milyaev et al., 2022*). Different phytohormones promote or inhibit flower bud differentiation. Plants transition from vegetative growth to reproductive growth by regulating the balance of various hormones (*Iqbal et al., 2017*). Previous studies postulate that endogenous tZR content increases progressively during flowering (*Pang et al., 2014*). Disrupted gibberellin biosynthesis and signal transduction lead to delayed flowering in GA mutant plants (*Lin et al., 2012*), while exogenous BR application delays flowering in wild-type Arabidopsis (*Noh et al., 2004*). Jasmonic acid (JA) is a lipid-derived phytohormone that participates in root elongation, petal growth, stamen development, senescence promotion, and stress response (*Sheng, 2020*). JA enhances GA signaling pathways to promote floral organ formation (*Hou*

*et al., 2010*). Endogenous hormones such as IAA, JA, and SA positively regulate bud germination and floral morphogenesis in kiwi fruit. In contrast, elevated tZR and ABA levels combined with reduced GA1 and SAG content facilitate leaf-to-flower bud transition (*Zhang, 2024*). Exogenous SA application induces endogenous SA synthesis and suppresses the expression of transcription factor FLC in wild-type Arabidopsis, thereby promoting flowering (*Khurana & Cleland, 1992*). Herein, the T2 group exhibited larger and fuller flower buds with more layered structures in longitudinal sections compared to the CK2 group. The increased JA content suggested its potential role in promoting flower bud differentiation.

The phenylpropanoid biosynthesis (map00940) pathway is crucial for flower bud differentiation (*Yao et al., 2021*). In this study, it was associated with biosynthesis of various plant secondary metabolites (map00999), flavonoid biosynthesis (map00941), and biosynthesis of various alkaloids (map00996). Biosynthesis of various plant secondary metabolites (map00999) included coumarin biosynthesis. Coumarins are crucial in keeping the structural integrity and signal transduction during flower bud differentiation (*Fan et al., 2018*). Notably, both T1 *vs* CK1 and T2 *vs* CK2 comparison groups exhibited parallel increases in coumarin and eriodictyol concentrations, suggesting potential functional synergy between these metabolites in bud differentiation. The T1 group exhibited elevated levels of 2-isopropylmalic acid, (+)-catechin, and trans-ferulic acid, indicative of their positive regulatory roles. Conversely, 3′, 5′-dimethoxy-3,5,7,4′-tetrahydroxyflavone and 4-hydroxyphenylpyruvic acid exhibited a decrease in concentration in these comparisons. In the T2 *vs* CK2 group, myricetin and isopimpinellin demonstrated significant upregulation, while umbelliferone and trans-caffeic acid exhibited inverse concentration trends. These differential accumulation patterns highlight the temporal specificity of metabolic regulation during flower bud differentiation.

Biosynthesis of various alkaloids (map00996) was associated with plant hormone signal transduction (map04075) and alpha-linolenic acid metabolism (map00592). Biosynthesis of various plant secondary metabolites (map00999) was associated with citrate cycle (map00020), carbon metabolism (map01200), which contained pentose phosphate pathway (map00030), and citrate cycle (map00020). The pentose phosphate pathway was associated with histidine metabolism (map00340). Notably, jasmonic acid and 6-phosphogluconic acid were significantly increased in T2 *vs* CK2. In contrast, the comparison group exhibited a decrease in trans-traumatic acid levels. These findings collectively reveal dynamic metabolic reprogramming events during bud differentiation.

The observed metabolic shifts and pathway-specific activation/inhibition signatures suggested a complex regulatory network of secondary metabolism to drive floral developmental transitions.

## CONCLUSIONS

This study combined physiological and metabolomics techniques to analyze nutrients and regulatory metabolites involved in flower bud differentiation post-thinning and reshaping of apple orchards. Thinning and reshaping had a direct effect on light penetration, which

resulted in higher levels of sucrose, glucose, fructose, and sorbitol. Moreover, flower bud differentiation during the early and late stages was promoted by the accumulation of coumarin (C05851), eriodictyol (C05631), and histidine (C00135).

The T1 group exhibited significant accumulation of 2-isopropylmalic acid (C02504), (+)-catechin (C09727), and trans-ferulic acid (C01494), suggesting their potential roles as positive regulators of metabolic activity. In contrast, 3′, 5′-dimethoxy-3, 5, 7, 4′-tetrahydroxyflavone (C11620) and 4-hydroxyphenylpyruvic acid (C01179) exhibited reduced concentrations across the comparison groups, implying distinct regulatory mechanisms. Temporal specificity in metabolic modulation was further evident in the T2 *vs* CK2 comparison. Myricetin (C10107) and isopimpinellin (C02162) exhibited significant upregulation, while umbelliferone (C09315) and trans-caffeic acid (C01197) concentrations exhibited opposing trends. Notably, there was a substantial increase in jasmonic acid (C08491) and 6-phosphogluconic acid (C00345) levels, accompanied by a concurrent decline in trans-traumatic acid (C16308) in the T2 *vs* CK2 comparison group. These shifts underscored the dynamic metabolic reprogramming events occurring during floral bud differentiation, with time-dependent modulation of pathway activities. However, the current evidence was insufficient to develop a clear regulatory relationship network. Future studies should consider setting up more comprehensive and precise observations and detections for several consecutive years and conducting experimental validation of targeted metabolites.

### Funding

This work was supported by the National Natural Science Foundation of China (Grants 32160683 and 31760556) and the Gansu Provincial Technology Innovation Guidance Program (Grant 22CX8NA026). The funders had no role in study design, data collection and analysis, decision to publish, or preparation of the manuscript.

### Grant Disclosures

The following grant information was disclosed by the authors:
Natural Science Foundation of China: 32160683, 31760556.
Gansu Provincial Technology Innovation Guidance Program: 22CX8NA026.

### Competing Interests

The authors declare there are no competing interests.

### Author Contributions

- Zehua Yang conceived and designed the experiments, performed the experiments, analyzed the data, prepared figures and/or tables, authored or reviewed drafts of the article, and approved the final draft.
- Tianli Guo conceived and designed the experiments, authored or reviewed drafts of the article, and approved the final draft.

- Junqiang Niu conceived and designed the experiments, performed the experiments, analyzed the data, authored or reviewed drafts of the article, and approved the final draft.
- Xiaoning Yin conceived and designed the experiments, performed the experiments, authored or reviewed drafts of the article, and approved the final draft.
- Ming Ma conceived and designed the experiments, performed the experiments, authored or reviewed drafts of the article, and approved the final draft.

## Data Availability

All untargeted metabolomic data used in this publication are available at EMBL-EBl MetaboLights: MTBLS11951.

The raw data is available in the Supplementary Files.

## Supplemental Information

Supplemental information for this article can be found online at http://dx.doi.org/10.7717/peerj.20011#supplemental-information.

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
