# Peer review of "Metabolomic and physiological analysis of bud differentiation in dense apple (Malus×domestica Borkh.) orchards following thinning and reshaping"

_PeerJ, doi:10.7717/peerj.20011_

## Round 0.1 · original submission · Major Revisions

The manuscript was reviewed by two independent experts in the field. Both the reviewers found the work interesting but raised several issues which should be addressed for further consideration. The reviewers provide detailed comments in their reviews and point out the areas where the manuscript needs to be improved.

Reviewer 1 ·

Basic reporting

Figure 3: It’s a bit hard to interpret the heatmap here, and whether there were any interesting results to glean from it. Is this actually discussed in the manuscript anywhere? What about all of the metabolites identified in the non-targeted analysis? The legend for A) does not seem to correlate with the pie chart. How were the metabolites characterised into the different compound classes? Were there any highly differential metabolites from the non-targeted analysis that are worth discussing or considering further as alternatives to the flavonoids signalled out from the targeted analysis?

Figure 5 is blurry and hard to read.

Line 86-87: Secondary metabolites, by their definition, are non-essential for growth and development. The authors should choose relevant, ideally more recent, literature references that adequately support their discussion of these flavonols. For instance, the cited review by Winkel-Shirley (2001) cites a study whereby flavonoid-deficient mutants of Arabidopsis led to higher susceptibility to UV radiation, but not death. Hence, that alone does not make these metabolites essential.

Experimental design

Methods and Materials: Apart from listing the instrument for which the LC-MS data was obtained, there are no details about the acquisition method, gradient, column used, etc., and these will need to be provided so that the experiment is repeatable and valid. Furthermore, there is no information as to how the metabolite annotation of the 845 metabolites was conducted, or how the non-targeted analysis was conducted. Was this by comparison of retention time to authentic standards run on the same column? Was MS/MS data collected and spectral matching was used for level 2 confident annotations? Or use of MRMs?

Line 160: How were metabolomics data normalised? Was this done before or within the MetaboAnalyst tool? Why were two different versions of MetaboAnalyst used (v5.0 referenced in line 167)?

Validity of the findings

Lines 350-351: “This finding led to a hypothesis that high eriodictyol concentration inhibits the differentiation of flower buds. thinning and reshaping.” Do you mean to say it “promotes” as the hypothesis provided here seems to contradict everything argued up until this point?

Lines 358-359: “The phenylpropanoid biosynthesis pathway is crucial for flower bud differentiation (Yao et al., 2021).” This requires more detail to convince me and the reader of what exactly makes this biosynthetic pathway crucial. It would be useful to see how the results here line up with the biochemistry. Are all metabolites in this pathway upregulated, or just some?

It would be useful to see a discussion on the potential photoprotective properties of these flavonoids as “sunscreens” and whether that is a contributing factor in the flower bud differentiation process.

Figure 6: Since these are relative abundances, and not concentrations, it should be stated in the caption here what the y-axis corresponds to (peak area, peak intensity, etc.). If normalisation has not been performed, these should be adjusted and reanalysed.

Additional comments

Lines 156-157: Delete ”, producing a significant amount of mass spectrometry data.” A superfluous statement.

·

Basic reporting

This study provides valuable insights into how thinning and reshaping influence bud differentiation in apple orchards through physiological and metabolic changes. The metabolomic analysis, revealing 845 metabolites, is particularly impressive, but the discussion could be strengthened by providing a more detailed mechanistic explanation of how specific compounds drive bud differentiation. While the study highlights phenylpropanoid and flavonoid biosynthesis pathways, further experimental validation of the roles of key metabolites such as coumarins and eriodictyol would enhance scientific rigor. A deeper discussion on how these metabolites interact with hormonal or signaling pathways would provide clearer causal links. Additionally, exploring functional assays or genetic studies could help confirm the biological relevance of these compounds. Given the significant metabolic changes observed, the study would benefit from a broader discussion of their physiological implications beyond sugar accumulation. While thinning improves light conditions, a more systematic approach to linking environmental changes with metabolic responses would clarify the findings. Overall, this work is impactful, but refining the discussion with more experimental and mechanistic insights will strengthen the study.

Experimental design

Line 125, it is unclear how the single factor was considered. Was it chosen randomly, or was there a specific criterion for selection? A more detailed explanation is necessary to ensure reproducibility.

Line 138, the classification of medium to short roots requires precise measurements; what was the exact length in cm used for differentiation?
Additionally, the choice of MetaboAnalyst versions raises questions. Why were versions 3 and 5 used instead of relying on a single version? Furthermore, given that MetaboAnalyst 6 is freely available online, what was the rationale for using older versions?

Experimental validation of the roles of key metabolites such as coumarins and eriodictyol would enhance scientific rigor.

Validity of the findings

Provide recent references for lines 319 and 322 to support your claims.
Line 382, include future applications of your findings and suggest experiments that could further enhance fruit quality using the observed metabolic changes.
Add a diagram summarizing all the metabolic pathways involved in thinning, illustrating their roles in bud differentiation. The diagram should clearly depict how these pathways contribute to the observed physiological and metabolic changes, strengthening the discussion.

Additional comments

Line 45, include a recent reference.
After line 85, ensure a smooth transition when discussing carbohydrates and flavonoids, establishing a clear link rather than shifting abruptly between topics. Additionally, when introducing flavonoids, integrate them naturally into the discussion instead of mentioning flavonoids, introduce polyphenols, and then flavonoids.
Line 113 could be rewritten in a clearer and more structured way to enhance the value of the work.

---

## Round 0.2 · Minor Revisions

Thanks for revising the manuscript. Some of the issues raised by reviewer # 2 is still unaddressed. Please respond to them in addition to minor comments raised by reviewer 1.

Reviewer 1 ·

Basic reporting

Just some minor fixes that should be made to the english used:
Line 124: Change "This study will elucidates" to "This study elucidates".
Line 192: Change "Versio 1.56.0" to "Version 1.56.0".
Line 429: Change "was relate to histidine: to "was related to histidine".

Experimental design

The methodologies have now been sufficiently described and are of a high quality for replication if needed.

Validity of the findings

All analyses are now of a very high standard.

Additional comments

This study investigated the physiological and metabolic impacts of thinning and reshaping on flower bud differentiation in dense apple orchards. The research found that thinning and reshaping significantly increased sugar content and altered the levels of various metabolites, including promoting compounds like coumarin and histidine during late flower bud differentiation, suggesting a key role in their development. These metabolic changes suggest a complex regulatory network as a response to the improved light penetration caused by thinning and reshaping, ultimately enhancing flower bud differentiation. This paper's detailed metabolomic analyses have been greatly imrpoved since the first iteration and offer a robust foundation for future studies exploring targeted interventions to optimize apple orchard productivity. Overall, it is my opinion, that the paper will be of relevance to a broad suite of researchers in the fields of agriculture, metabolomics and chemical ecology.

---

## Round 0.3 · Minor Revisions

Thank you for revising the manuscript. While the scientific rigor has significantly improved, the quality of the English language is still not suitable for publication in PeerJ. I strongly encourage the authors to have the manuscript revised by a proficient English speaker or a professional language editing service.

**Language Note:** The Academic Editor has identified that the English language must be improved. PeerJ can provide language editing services - please contact us at [email protected] for pricing (be sure to provide your manuscript number and title). Alternatively, you should make your own arrangements to improve the language quality and provide details in your response letter. – PeerJ Staff

---

## Round 0.4 · accepted · Accept

I thank the authors for revising the manuscript. The current version is satisfactory for publication.